# The Widening Gap between the Digital Capability of the Care Workforce and Technology-Enabled Healthcare Delivery: A Nursing and Allied Health Analysis

**DOI:** 10.3390/healthcare11070994

**Published:** 2023-03-30

**Authors:** Meg E. Morris, Natasha K. Brusco, Jeff Jones, Nicholas F. Taylor, Christine E. East, Adam I. Semciw, Kristina Edvardsson, Claire Thwaites, Sharon L. Bourke, Urooj Raza Khan, Sally Fowler-Davis, Brian Oldenburg

**Affiliations:** 1ARCH, School of Allied Health, Human Services and Sport, La Trobe University, Bundoora, VIC 3086, Australia; n.taylor@latrobe.edu.au (N.F.T.); a.semciw@latrobe.edu.au (A.I.S.); c.thwaites@latrobe.edu.au (C.T.); 2Victorian Rehabilitation Centre, Healthscope, Melbourne, VIC 3150, Australia; 3School of Allied Health, Human Services and Sport, La Trobe University, Bundoora, VIC 3086, Australia; natasha.brusco@monash.edu; 4Rehabilitation, Ageing and Independent Living (RAIL) Research Centre, School of Primary and Allied Health Care, Monash University, Frankston, VIC 3199, Australia; 5Digital Innovation Hub, Cisco Innovation Central Melbourne, Office of the DVC Research and Industry Engagement, La Trobe University, Bundoora, VIC 3086, Australia; 6Eastern Health, Box Hill, VIC 3128, Australia; 7ARCH, School of Nursing and Midwifery, La Trobe University, Bundoora, VIC 3086, Australia; c.east@latrobe.edu.au (C.E.E.); k.edvardsson@latrobe.edu.au (K.E.); s.bourke@latrobe.edu.au (S.L.B.); 8Mercy Health, Heidelberg, VIC 3084, Australia; 9Department of Allied Health, Northern Health, Epping, VIC 3076, Australia; 10School of Psychology and Public Health, La Trobe University, Bundoora, VIC 3086, Australia; 11Advanced Wellbeing Research Centre, Sheffield Hallam University, Sheffield S10 2BP, UK; s.fowler-davis@shu.ac.uk; 12ARCH, School of Psychology and Public Health, La Trobe University, Bundoora, VIC 3086, Australia; b.oldenburg@latrobe.edu.au; 13Baker Heart and Diabetes Institute, Melbourne, VIC 3004, Australia; 14Alfred Health, Melbourne, VIC 3004, Australia

**Keywords:** digital, digital maturity, digital capability, digital literacy, healthcare, nursing, midwifery, allied health, workforce, consumer, care worker, care economy

## Abstract

There is a need to ensure that healthcare organisations enable their workforces to use digital methods in service delivery. This study aimed to evaluate the current level of digital understanding and ability in nursing, midwifery, and allied health workforces and identify some of the training requirements to improve digital literacy in these health professionals. Representatives from eight healthcare organizations in Victoria, Australia participated in focus groups. Three digital frameworks informed the focus group topic guide that sought to examine the barriers and enablers to adopting digital healthcare along with training requirements to improve digital literacy. Twenty-three participants self-rated digital knowledge and skills using Likert scales and attended the focus groups. Mid-range scores were given for digital ability in nursing, midwifery, and allied health professionals. Focus group participants expressed concern over the gap between their organizations’ adoption of digital methods relative to their digital ability, and there were concerns about cyber security. Participants also saw a need for the inclusion of consumers in digital design. Given the widening gap between digital innovation and health workforce digital capability, there is a need to accelerate digital literacy by rapidly deploying education and training and policies and procedures for digital service delivery.

## 1. Introduction

Digital healthcare, digital technologies, and digital systems have the potential to rapidly transform the care workforce design and care delivery [1,2,3] across the care economy [4,5,6]. Digital healthcare can also improve equity and access to services for people in regional, remote, and rural communities [7] and for diverse cultures and health needs [8]. However, innovations and technology use in health services are rarely assessed for impact, creating the risk of poor adoption of best practice digital healthcare [9].

For the care workforce, digital maturity, literacy, and capabilities respectively refer to (i) “the extent to which its health information technology is an enabler of high-quality care through supporting improvements to service delivery and patient experience” [10], (ii) “the mastery of simple and practical skills which bring a profound enrichment and transformation of human thinking capabilities” [11,12], and (iii) “the ability to live, work, participate and thrive in a digital world” [13] and this requires “particular knowledge and attitudes regarding legal and ethical aspects, privacy and security, as well as understanding the role of information communication technology in society” [14].

One of the greatest challenges to the care economy since the COVID-19 pandemic has been to ensure the health workforce has sufficient capability, knowledge, and skills to deliver digitally enabled healthcare delivery [15,16,17,18]. While technology provides the opportunity for improved health care, health workers are essential for implementing the technology into productive use [6,19,20]. Yet, the availability of digital health technologies has arguably accelerated beyond current levels of care workforce digital capability [13,21]. For example, there have been rapid advances in “online” emergency department triaging, mobile health applications, wearable sensors for remote monitoring, smart homes, and artificial intelligence and machine learning for predictive risk stratification [22]. In addition to building workforce digital capability, there is a need for health organisations to adopt culture, policies, and clinical practices to ensure they are innovation ready [9].

There have been more than 30 published digital health competency frameworks for the health workforce [23]. From these, 28 domains have been identified, such as basic information technology literacy, health information management, digital communication, ethical, legal, and regulatory requirements for digital healthcare, and data privacy and security [23]. There is a need for future digital health training to focus on competencies relevant to a particular health care worker group, role, level of seniority, and setting [23]. Consumer co-design and co-production of digital health innovations also warrant inclusion [24,25,26,27].

In 2020–2021 in Australia, three key digital reports were published with domains to guide future work [15,19,28]. Informed by these reports, this study aimed to answer the following research questions: what is the current level of digital maturity and digital capability across nursing, midwifery, and allied health workforces in Australian healthcare settings? What are the training requirements needed to improve health professional digital literacy?

## 2. Materials and Methods

Ethical considerations and reporting standards: This study was approved by the La Trobe University Human Research Ethics Committee (REF HEC22050) and reported according to the COnsolidated criteria for REporting Qualitative research (COREQ) checklist [29] (Appendix A). All participants provided written informed consent prior to participating in the focus groups.

Setting: Eight public/private health services across Victoria, Australia participated in this study. These services provide health care to 42% of the Victorian population and 11% of the Australian population.

Participants: Purposeful sampling aimed to recruit three participants from each health service. This included the university academic leader, Director of Nursing and Midwifery and Director of Allied Health, or proxies, from each health service (projected sample size *n* = 24). Participants were approached via an email from an independent researcher (NB).

Data collection: Data were collected in July 2022 in two stages. The first stage was a 1 h group seminar for all participants. This seminar had an education focus and introduced three key frameworks: Victoria’s Digital Health Roadmap 2021–2025 [19], the National Nursing and Midwifery Digital Health Capability Framework [15], and the Digital Health Capability Framework for Allied Health Professionals [28]. The second stage was the focus groups [29,30] (see checklist in Appendix A for details).

Inclusion Criteria: Staff employed at one of the eight participating health services as the Director of Nursing and Midwifery, Director of Allied Health; or employed at La Trobe University, or their proxies. Exclusion: No specific exclusion criteria.

### 2.1. Self-Reported Levels of Digital Maturity and Capability

Analysis: To assess the current level of organisational digital maturity and capability, focus groups participants applied the rating system described in the three frameworks [15,19,28] to different aspects of their organisation (Figure 1, Figure 2 and Figure 3 note the different domains and Likert scales for the three frameworks). The ratings reported during the focus group were tabulated deductively and presented as a count for each domain of the three frameworks.

### 2.2. Themes from the Focus Groups

Analysis: To understand participant experiences of digital technology in healthcare, an inductive thematic analysis was undertaken [31]. Based on the thematic analysis framework developed by Braun and Clarke [31,32,33], two researchers familiarised themselves with the data while facilitating the focus groups and then by re-reading the full transcripts; generated initial codes; searched, reviewed, defined, and named the themes; and produced the study findings. While the interview guide for the focus groups were structured around three key frameworks [15,19,28], the initial codes and themes were generated independently of these frameworks.

### 2.3. Training Requirements to Improve Digital Literacy

Analysis: Participant comments relating to training requirements needed to improve healthcare professional digital literacy were reported alongside the thematic analysis.

## 3. Results

Across the 8 focus groups, there were 23 participants and 17 were female. This included seven University Academic Leads, seven Directors of Nursing and Midwifery, and nine Directors of Allied Health, or proxies, all with 7+ years of experience in the healthcare or university setting. Focus group size ranged from two to four. Individual health service summaries were made available to participants for the member checking process with three participants providing feedback and modifications.

### 3.1. Self-Reported Levels of Digital Maturity and Capability

Based on Victoria’s Digital Health Roadmap 2021–2025 [19], the majority of self-ratings of digital maturity was developing (level 3 of 5) with no level 5 transformative ratings (Figure 1). Based on the National Nursing and Midwifery Digital Health Capability Framework [15], nursing and midwifery mostly identified an intermediate level (level 2 of 5) of capability relating to digital professionalism, leadership, and advocacy (Figure 2). Based on the Digital Health Capability Framework for Allied Health Professionals [28], self-rated allied health digital capability was most commonly identified at a consolidated level (level 2 of 4) of capability relating to the digital workplace and digital professionalism (Figure 3).

### 3.2. Themes from the Focus Groups

Theme 1. Participants reported a gap between the rapid growth in the deployment of digital healthcare technologies and systems and the digital capability of the workforce. The pandemic was reported to exacerbate this gap, “with the onset of COVID, the rollout of telehealth was rapid, and it became a substitution model” (Allied Health, Male). Communication strategies need to consider varied access to digital systems with a participant stating, “It’s always difficult trying to get a message across 11,000 staff, as not everyone is on electronic email” (Allied Health, Female).

Theme 2. The theme of ‘Nothing about us without us’ [34] was expressed across a number of health services who highlighted the need “to get more consumer engagement in digital innovations” (University Academic Lead, Female) and that there should be “a big focus on the consumer and knowing your consumer intimately” (Allied Health, Male). In addition, consumer access to their own information was seen as essential, with several of the health services providing patient portals which had variable levels of functionality, ranging from basic to modest.

Theme 3. Cyber safety was frequently discussed and all health services referred to the cyber-attack on a public Victorian health service in 2021 [35], as well as ensuring the secure management of digital data. Following this cyber-attack there was a range of responses to address cyber security across the health services, including both infrastructure upgrade and staff training. Responses ranged from reactive, “It’s more of a reactive approach to cybersecurity and noncompliance with our security controls” (Allied Health, Female) to proactive, “there is a lot that’s in place to help improve Cyber Security since the 21 cyber-attack on [removed for anonymity] Health. Our cybersecurity department had one personnel [prior to 2021], now we have 10 people in that team. There is also a strategic pillar dedicating digital health strategy to cybersecurity” (Nursing and Midwifery, Male).

Theme 4. Joining up health care records varied within and between health services, and this was aligned to the electronic medical record (EMR) maturity for the health service. Across the health services, EMR maturity ranged from no EMR to a well-established EMR. In general, the EMR was reported to reduce duplication. However, use of the EMR at the bedside was reported to sometimes reduce connections and time spent with patients. Collaboration between hospitals to share EMRs was seen as a positive step to enable seamless care when patients presented to multiple health services over time.

Theme 5. In relation to digital healthcare, the term ‘pain point’ was used by multiple participants and this referred to digital systems that do not talk to each other, differing levels of digital maturity across a single health service, and healthcare records that do not connect. Logging into multiple systems was cumbersome with one participant reporting that “we went from 15 logins to one… that is so much better than it used to be and much safer.” (Allied Health, Female). Participants also reported that staff “… get really frustrated. So in the end, they just ring up people instead of going digital, … as they can’t remember their password.” (Nursing and Midwifery, Female).

Theme 6. Participants highlighted that adequate access to digital infrastructure is essential to the digital capability of the workforce. “You’ve got the infrastructure enablers, and then you’ve got the capability enablers. Both considerations are important, it is about identifying structural things that need to be in place, and you’re talking about Internet, access to equipment and things like that.” (Nursing and Midwifery, Female). Other participants reported that for staff “it’s hard for them to access to reliable internet.” (Allied Health, Male), and “the issue with nursing is access to the computer” (Nursing and Midwifery, Female). Participants noted that they were not aware of all the digital infrastructure available, with one asking the focus group “What are all our current digital platforms?” (Nurse, Female).

Theme 7. Virtual health was rapidly implemented during the COVID-19 pandemic. “I think what our clinicians are really wanting is some guidance around when to use virtual health, when not to use it, and also how to ensure the patients have the digital health literacy… We can say it’s clinically indicated, but the patient might not have the tools and resources to do it” (Allied Health, Female) and “Virtual health … is still a bit clunky, in lots of ways. And particularly because this was, in our context, implemented in the middle of [the] pandemic. That was a challenging environment to completely restructure the way you work” (Allied Health, Female). Another reported that “Telehealth was not particularly intuitive; it was difficult to use. So, there were easier ways, such as a phone call.” (Nursing and Midwifery, Male). It was also noted that “We need to have insight from a vulnerable consumer perspective, I’ve not heard conversations happening here about the use of telehealth and patients that are so economically stretched, that the thought of being offered a phone call or video session, is probably stressing them out because they don’t have the data, they can’t afford the phone bill or the electricity to charge the phone, if they’re lucky enough to even have a smartphone. In addition, we have got a huge multicultural community that we support, where most of the platforms are in English.” (Allied Health, Female).

Theme 8. There was variable digital capability across the nursing and midwifery workforce. “We have a bunch of people that are really excited and can see that this is going to have a huge effect on consumers as well as data … And I think we have another group that are scared and fearful. And I don’t necessarily think that fits into age groups. I think it’s about people and a fear of the unknown” (Nursing and Midwifery, Female). Another participant reported that “Digital is not about digital, but it’s actually the gateway to data. And so it’s about thinking about how people actually conceive what data can do for them. For the nursing workforce, that’s what you do every day, you take patient data and synthesise it and act on it. And that ‘data mindset’ actually requires different skill sets and training.” (Nursing and Midwifery).

### 3.3. Training Requirements to Improve Digital Literacy

Participants reported that to improve health care staff digital literacy, health services are currently developing solutions in silos and suggested a state-based or national approach may be more efficient. Some advised that not all staff need to be digital experts and that having a small number of digital experts across the workforce would ensure staff have access to in-the-moment training. Figure 4 summarises the ten strategies that emerged from the focus groups, which could improve the digital literacy of healthcare professionals.

## 4. Discussion

Digital maturity and capability were highly variable within the health professions, as well as between Australian health services. Participants reported an increasing gap between the rate of growth in digital healthcare technologies, systems, and data and the comparatively low level of digital literacy and capability in the workforce. They also highlighted the need for consumers to be involved in digital healthcare delivery design. Training requirements to improve digital literacy focused on the minimum standard for digital literacy, training for different career stages and on the use of data, and training to keep up with digital innovation. A recurring theme was the need for health professional training to improve digital literacy and capability in delivering virtual care, as well as ensuring secure management of digital data.

Facilitation and familiarisation with digital technologies and their implementation is linked to digital confidence and utility [36]. As identified in the current study, the adoption of digital health technologies faces many challenges, such as the segregation of digital systems within and between health services [5], low levels of basic information technology literacy in health professionals and managers [23], unreliable access to digital technology or lack of evidence supporting their use [36], and lack of mandatory digital health training on entry to care roles [23].

While Australian state and national frameworks have been developed for digital maturity, capability, education, and workforce [15,19,28,37], the participants in this study reported experiencing siloed and uncoordinated effort into the implementation of the frameworks. Workforce shortages impacted participation in digital education and training, and there was little access to digital micro-credentialling. Policy makers can use these findings to prioritise the greatest areas of need for digital technology and its subsequent implementation.

The key strengths of the current study are the representative sample of health services who provide health care to almost half of the Victorian population as well as the inclusion of multiple professions. In addition, the two-phase design of this study, that included the provision of a 1 h seminar prior to the 1 h focus group, means that all focus group participants were socialized to the three key frameworks (Victoria’s Digital Health Roadmap 2021–2025 [19], the National Nursing and Midwifery Digital Health Capability Framework [15], and the Digital Health Capability Framework for Allied Health Professionals [28]), as well as the study aims and design, prior to participating in the focus group. The main limitation was the comparatively modest number of key representatives from each of the health professions. As the representative was assigned to participate, there was an assumption that the views of the participants were representative of the health service they were aligned with. The other main limitation was that the focus groups did not include medical professionals. This means that the results have limited generalizability beyond allied health, nursing, and midwifery. Consumers were not included in the focus groups and there remains a need for future trials to capture the views of the patients and families who are the recipients of digital healthcare.

## 5. Conclusions

To conclude, there is a gap between the digital capability of the current health workforce and the need for the rapid deployment of high-quality digital healthcare to patients with a wide range of health conditions. Leaders from allied health, nursing, and midwifery professions articulated clear strategies to improve the digital literacy of healthcare professionals. These await consistent implementation nationally and globally.

## Figures and Tables

**Figure 1 healthcare-11-00994-f001:**
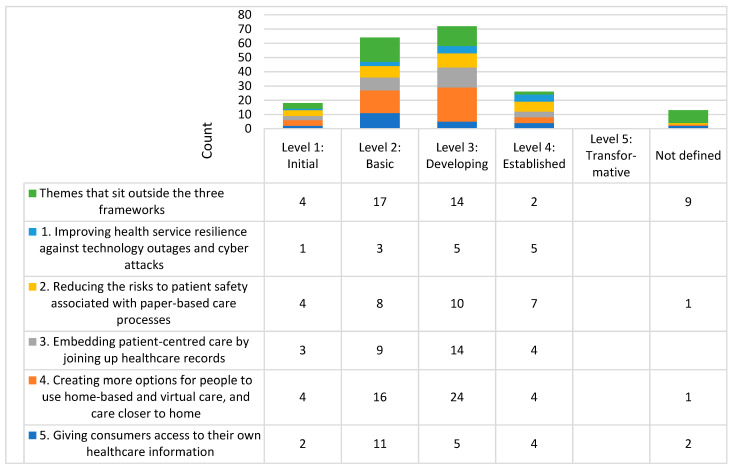
Histogram of self-rated digital maturity across the eight health services. NB each number represents the number of times (counts) the domain and level were mentioned in the focus groups.

**Figure 2 healthcare-11-00994-f002:**
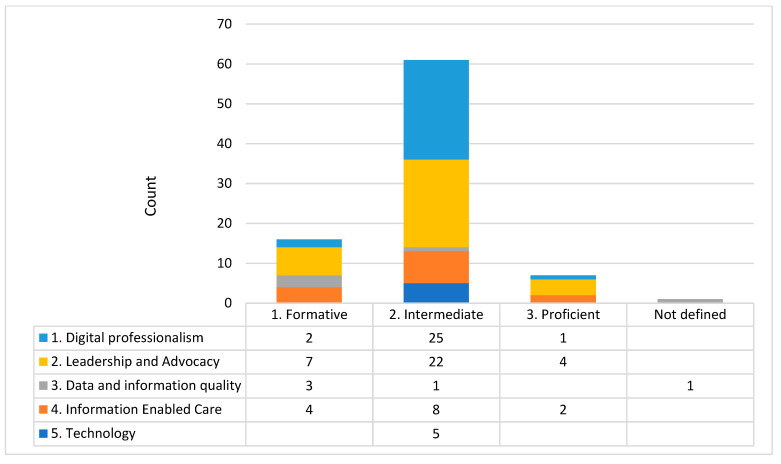
Histogram of self-rated digital capability for the nursing and midwifery workforce. NB each number represents the number of times (counts) the domain and level were mentioned in the focus groups.

**Figure 3 healthcare-11-00994-f003:**
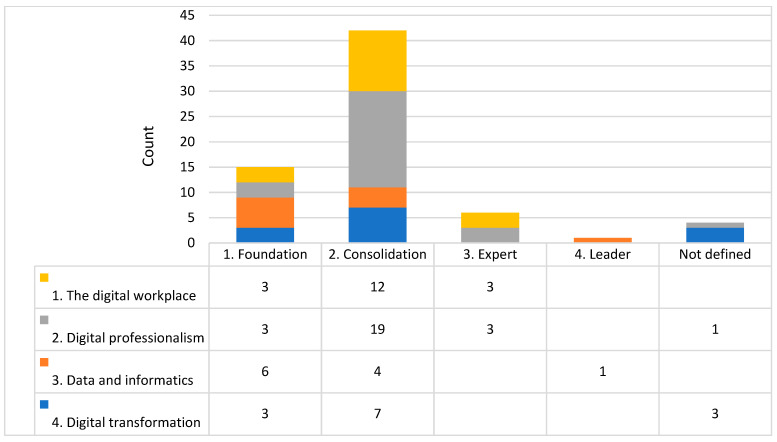
Histogram of self-rated digital capability for the allied health workforce. NB each number represents the number of times (counts) the domain and level were mentioned in the focus groups.

**Figure 4 healthcare-11-00994-f004:**
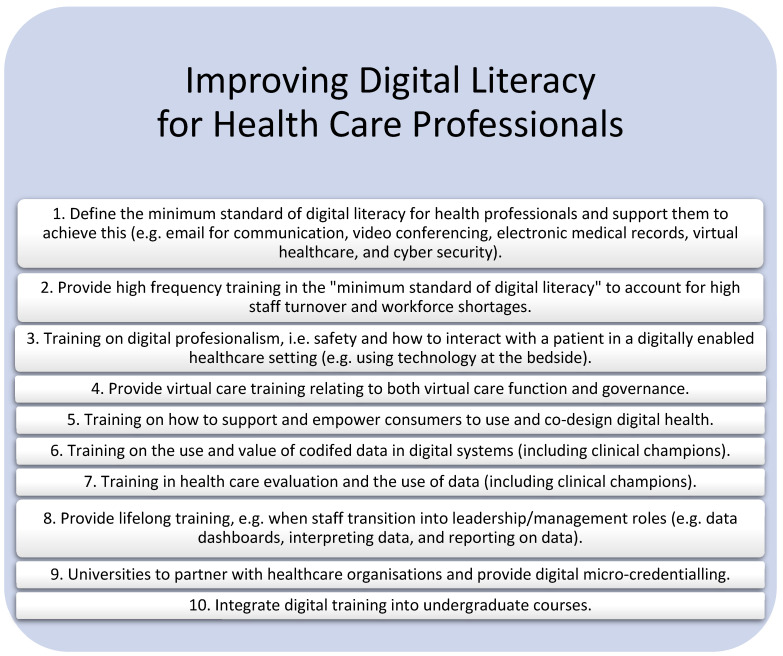
Strategies to improve the digital literacy of healthcare professionals.

## Data Availability

The data that support this study cannot be publicly shared due to ethical or privacy reasons and may be shared upon reasonable request to the corresponding author if appropriate.

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
