# Peer review of "The Widening Gap between the Digital Capability of the Care Workforce and Technology-Enabled Healthcare Delivery: A Nursing and Allied Health Analysis"

_healthcare, 2023, doi:10.3390/healthcare11070994_

Round 1

Reviewer 1 Report

Thanks for the possibililty to review this manuscript, I have only some minor formal issues to comment:

·        Please provide a clear research question

·        Please provide a reference for the focus group method

·        Please provide a table with the main characteristics of study particpants
(e.g. position, experience, gender, education etc.)

·        Line 190: Please correct “of the pandemic” instead of “of a pandemic”

·        Please check for double space characters (e.g. line 191, line 208, appendix 1 throughout etc.)

·        Please extend the limits and strengths section substantially

·        Appendix 1: Please capitalize all terms in the title that relate to the acronym COREQ

·        Appendix 1: Please check for any formal inconsistencies (double space characters, interpunctuation, capitalization of headings/subheadings)

·        Appendix 1: There is something wrong with the sentence in line 13

·        Table 1-3: Shouldn’t it be “… mentioned in the focus groups” instead of “… mentioned in the focus group”?

·        Figure 1: Font size of the title within the figure is somewhat to large

·        Please check reference list for any inconsistencies and errors

Author Response

Thanks for the possibility to review this manuscript, I have only some minor formal issues to comment:

  1. Please provide a clear research question

RESPONSE: The following has been added: “This study aimed to answer the following research questions; what is the current level of digital maturity and digital capability across nursing, midwifery and allied health workforces in Australian healthcare settings?; and what are the training requirements needed to improve health professional digital literacy?”

  1. Please provide a reference for the focus group method

RESPONSE: This has been added to the methods section.

  1. Please provide a table with the main characteristics of study participants
    (e.g. position, experience, gender, education etc.)

RESPONSE: Thank you for this suggestion. Our team has taken care to ensure that the main characteristics of the study participants are now adequately summarized in the first paragraph of the results section, and an additional table is therefore not required

  1. Line 190: Please correct “of the pandemic” instead of “of a pandemic”

RESPONSE: This has been corrected.

  1. Please check for double space characters (e.g. line 191, line 208, appendix 1 throughout etc.)

RESPONSE: Thankyou for picking this up, it has been corrected.

  1. Please extend the limits and strengths section substantially

RESPONSE: Thankyou for this suggestion. The following has been added to the strength / limitations section: “The key strengths of the current study are the representative sample of health services who provide health care to almost half of the Victorian population as well as the inclusion of multiple professions. In addition, the two-phase design of this study, that included the provision of a 1-hour seminar prior to the 1-hour focus group, means that all focus group participants were socialized to the three key frameworks three key frameworks (Victoria’s Digital Health Roadmap 2021–2025 [19], the National Nursing and Midwifery Digital Health Capability Framework [15] and the Digital Health Capability Framework for Allied Health Professionals [28]), as well as the study aims and design, prior to participating in the focus group. The main limitation was the comparatively modest number of key representatives from each of the health professions. As the representative assigned to participate there was an assumption that the views of the participants were representative of the health service they were aligned with. The other main limitation was that the focus groups did not include medical professionals. This means that the results have limited generalizability beyond allied health, nursing and midwifery. Consumers were not included in the focus groups and there remains a need for future trials to capture the views of the patients and families who are the recipients of digital healthcare.”

  1. Appendix 1: Please capitalize all terms in the title that relate to the acronym COREQ

RESPONSE: This has been corrected

  1. Appendix 1: Please check for any formal inconsistencies (double space characters, interpunctuation, capitalization of headings/subheadings)

RESPONSE: Thankyou, this has been corrected.

  1. Appendix 1: There is something wrong with the sentence in line 13

RESPONSE: The COREQ guide has been re-read and this section has been simplified according to the guide “No people refused to participate or dropped out.”

  1. Table 1-3: Shouldn’t it be “… mentioned in the focus groups” instead of “… mentioned in the focus group”?

RESPONSE: Thankyou, this has been corrected

  1. Figure 1: Font size of the title within the figure is somewhat to large

RESPONSE: We have reduced the font size

  1. Please check reference list for any inconsistencies and errors

RESPONSE: Thankyou we have checked and updated the reference list to ensure it is consistent and errors removed.

Reviewer 2 Report

The authors aimed to evaluate the current level of digital maturity and capability across the nursing, midwifery, and allied health workforces; and understand the training requirements to improve their digital literacy. Representatives from eight healthcare organisations within the Academic Research and Health Collaborative (ARCH) in Victoria, Australia, participated in one of eight focus groups (July 2022). Barriers and enablers to adopting digital healthcare and training requirements to improve digital literacy were discussed. Self-rated digital maturity and capability were reported using Likert scales.

The study faces a strategic them in the healthcare: the digital divide (DD).

The level of literacy is an important issue to investigate in the DD.

The ms needs some improvements before being considered for acceptance.

I have some comments for the authors:

1.       The abstract must be improved. It is now more a list of things. It must better summarize the sections with a clear structure.

2.       Where is the purpose? Insert the purpose with the main aim and the sub aims with bullet points.

3.       My idea is that the methods and results must be rearranged. Now they seem a list of (interesting) idea and reflections. Use editorial tools, paragraphs and tables. Results are arranged into themes. This is good however you must explain the choices.

4.       Insert the labels in the histograms of the figures (named  tables)?

5.       Expand the limitations in the discussion and explain them better.

6.       Conclusion is not a conclusion, it is more a solemn declaration. It must be derived by the results of the study and detailed better.

Author Response

The authors aimed to evaluate the current level of digital maturity and capability across the nursing, midwifery, and allied health workforces; and understand the training requirements to improve their digital literacy. Representatives from eight healthcare organisations within the Academic Research and Health Collaborative (ARCH) in Victoria, Australia, participated in one of eight focus groups (July 2022). Barriers and enablers to adopting digital healthcare and training requirements to improve digital literacy were discussed. Self-rated digital maturity and capability were reported using Likert scales.

The study faces a strategic theme in the healthcare: the digital divide (DD).

The level of literacy is an important issue to investigate in the DD.

The ms needs some improvements before being considered for acceptance.

I have some comments for the authors:

  1. The abstract must be improved. It is now more a list of things. It must better summarize the sections with a clear structure.

RESPONSE: Thankyou for this feedback. We have revised the abstract structure per the author guidelines. Based on your suggestion, it has been updated to include the following:

“There is a need to ensure that healthcare organizations enable their workforces to use digital methods in service delivery. This study aimed to evaluate the current level of digital understanding and ability in nursing, midwifery, and allied health workforces and identify some of the training requirements to improve digital literacy in these health professionals. Representatives from eight healthcare organizations in Victoria, Australia, participated in focus groups. Three digital frameworks informed the focus group topic guide that sought to examine the barriers and enablers to adopting digital healthcare along with training requirements to improve digital literacy. Twenty-three participants self-rated digital knowledge and skills using Likert scales and attended the focus groups. Mid-range scores were given for digital ability in nursing, midwifery, and allied health professionals. Focus group participants expressed concern over the gap between their organizations’ adoption of digital methods, relative to their digital ability and there were concerns about cyber security. Participants also saw a need for the inclusion of consumers in digital design. Given the widening gap between digital innovation and health workforce digital capability, there is a need to accelerate digital literacy by rapidly deploying education and training and policies and procedures for digital service delivery.”

  1. Where is the purpose? Insert the purpose with the main aim and the sub aims with bullet points.

RESPONSE: As per your comments as well as reviewer 1 comments, we have reframed the study with clear research questions. “In 2020-21 in Australia, three key digital reports were published with domains to guide future work [15, 19, 28]. Informed by these reports, this study aimed to answer the following research questions; what is the current level of digital maturity and digital capability across nursing, midwifery and allied health workforces in Australian healthcare settings?; and what are the training requirements needed to improve health professional digital literacy?”

  1. My idea is that the methods and results must be rearranged. Now they seem a list of (interesting) idea and reflections. Use editorial tools, paragraphs and tables. Results are arranged into themes. This is good however you must explain the choices.

RESPONSE: The structure of the method and result sections follow the author guidelines for this journal. Nevertheless, we have highlighted the sections with consistent sub-headings as well as highlighted the themes in the results section number.

  1. Insert the labels in the histograms of the figures (named  tables)?

RESPONSE: Thank-you for this suggestion, we have re-labelled these figures as histograms.

  1. Expand the limitations in the discussion and explain them better.

RESPONSE: Thank-you for this suggestion. The following has been added to the strength / limitations section:

“The key strengths of the current study are the representative sample of health services who provide health care to almost half of the Victorian population as well as the inclusion of multiple professions. In addition, the two-phase design of this study, that included the provision of a 1-hour seminar prior to the 1-hour focus group, means that all focus group participants were socialized to the three key frameworks three key frameworks (Victoria’s Digital Health Roadmap 2021–2025 [19], the National Nursing and Midwifery Digital Health Capability Framework [15] and the Digital Health Capability Framework for Allied Health Professionals [28]), as well as the study aims and design, prior to participating in the focus group. The main limitation was the comparatively modest number of key representatives from each of the health professions. As the representative assigned to participate there was an assumption that the views of the participants were representative of the health service they were aligned with. The other main limitation was that the focus groups did not include medical professionals. This means that the results have limited generalizability beyond allied health, nursing and midwifery. Consumers were not included in the focus groups and there remains a need for future trials to capture the views of the patients and families who are the recipients of digital healthcare .”

  1. Conclusion is not a conclusion, it is more a solemn declaration. It must be derived by the results of the study and detailed better.

RESPONSE: Thank you for pointing this out. We agree with the reviewer and have revised the conclusion accordingly:

“To conclude, there is a gap between the digital capability of the current health workforce and the need for rapid deployment of high quality digital healthcare to patients with a wide range of health conditions. Leaders from allied health, nursing and midwifery professions articulated clear strategies to improve the digital literacy of healthcare professionals. These await consistent implementation nationally and globally.”

Round 2

Reviewer 2 Report

N/A